# Comparison of Dual Beam Dispersive and FTNIR Spectroscopy for Lactate Detection

**DOI:** 10.3390/s21051891

**Published:** 2021-03-08

**Authors:** Nystha Baishya, Mohammad Mamouei, Karthik Budidha, Meha Qassem, Pankaj Vadgama, Panayiotis A. Kyriacou

**Affiliations:** 1Research Centre for Biomedical Engineering, School of Engineering and Mathematical Sciences, University of London, Northampton Square, London EC1V 0HB, UK; Mohammad.mamouei.2@city.ac.uk (M.M.); karthik.budidha@city.ac.uk (K.B.); Meha.Qassem@city.ac.uk (M.Q.); P.Kyriacou@city.ac.uk (P.A.K.); 2Interdisciplinary Research Centre (IRC) in Biomedical Materials, Queen Mary University of London, Mile End Road, London E1 4NS, UK; p.vadgama@qmul.ac.uk

**Keywords:** near infrared spectroscopy, dual beam dispersive NIR, FTNIR, lactate, aquaphotomics

## Abstract

Near Infrared (800–2500 nm) spectroscopy has been extensively used in biomedical applications, as it offers rapid, in vivo, bed-side monitoring of important haemodynamic parameters, which is especially important in critical care settings. However, the choice of NIR spectrometer needs to be investigated for biomedical applications, as both the dual beam dispersive spectrophotomer and the FTNIR spectrometer have their own advantages and disadvantages. In this study, predictive analysis of lactate concentrations in whole blood were undertaken using multivariate techniques on spectra obtained from the two spectrometer types simultaneously and results were compared. Results showed significant improvement in predicting analyte concentration when analysis was performed on full range spectral data. This is in comparison to analysis of limited spectral regions or lactate signature peaks, which yielded poorer prediction models. Furthermore, for the same region, FTNIR showed 10% better predictive capability than the dual beam dispersive NIR spectrometer.

## 1. Introduction

Near Infrared Spectroscopy, a type of molecular vibrational spectroscopy, has found immense applications in healthcare monitoring because at the wavelength range of 800–1200 nm, electromagnetic radiation has the ability to penetrate tissue and be retrieved in-vivo, making it an ideal non-invasive technique [1]. Another attractive feature of this technique, especially for intensive care, is its ability to perform real time, bed side monitoring using portable instruments [2]. These fundamental advantages of NIR spectroscopy have stimulated research for decades and established numerous applications, especially in extracting information such as, oxygenation level [3,4], glucose [5,6] and lactate [7] concentrations in blood, one of the most complex of biological matrices. NIR spectra of whole blood can now be acquired using a wide range of spectrometer types.

Spectrometers based on dispersive methods for acquiring NIR spectra have been available since the 1940s; the first economical commercial dual beam spectrophotometer (INFRACORD) was introduced by Perkin Elmer in 1957 [8]. It was not until 1969, however, that the first FTIR instrument (Model FTS-14) was introduced commercially by Digilab Inc. [9]. A dual beam dispersive spectrophotometer is more sensitive (with high Signal to Noise Ratio), provides better stability (able to subtract out the background noise without operator intervention), has a greater accuracy and better spectral resolution [10]. However, there are three major advantages of FTNIR over the archetype dual beam instrument:Multiplexing: Interferometers, which are an integral part of the FTIR/FTNIR spectrometers, use a moving mirror which can scan the entire range of wavelengths at once for spectra collection. This makes it faster than the dual beam spectrophotometer, which considers each wavelength, individually, in the spectral range for collection. Each of the scans in FTIR/FTNIR can be summated for a better accuracy and signal to noise ratio (SNR) (squareroot of the total spectra) for each spectrum, known as Fellgett’s advantage [11]Throughput: A FTIR/FTNIR spectrometer has a higher throughput compared to the dual beam spectrophotometer, owing to the avoidance of the slits or optical apertures and the use of fewer mirrors in the instrument, which increases the total energy of the incident light on a sample and reduces reflection losses. This increases SNR, and is known as Jacquinot’s advantage [12].Precision: The use of laser light sources to control the speed of mirror movement and instrument calibration makes FTIR/FTNIR capable of acquiring precise absorption spectra, independent of time or any other external variables, such as, temperature and vibrational variation [13,14].

From this, one instrument type could still be favoured over the other, but there have been no studies yet to formally compare the two, in terms of instrument/technology capabilities for biomedical applications in the NIR region. Similar studies has been performed for other fields of science, mostly, food [13,14,15,16,17,18,19]. However, studies related to biomedical sciences for the detection of biomarkers in blood has not been investigated previously. These biomarkers act as a reflection of personal health, and lactate is a key indicator of metabolic health for well bein [20] and haemodynamic shock states in critical care environments [21]. Elevated lactate levels or hyperlactaemia is an imperative biomarker for metabolic diseases like diabetes, hypertension and cardiovascular disease [22]. Hence, for the first time, spectra from whole blood sample solutions, (red cells, white cells, and platelets, suspended in plasma) with varying lactate concentrations, were acquired simultaneously from two representative instruments; the dual beam spectrophotometer, LAMBDA 1050 and the FTNIR/IR, Frontier${}^{\mathrm{TM}}$ both from Perkin Elmer (Waltham, MA, USA). The spectra were then analysed using the principles of Aquaphotomics and multivariate analysis techniques, and were compared for different regions of the NIR spectral range. In the regions of the NIR, where OH has a huge presence, Aquaphotomics principles were applied to understand the changes in the spectra due to lactate concentration changes. Multivariate analysis together with chemometric techniques have enabled decomposition and analysis of NIR spectroscopic data by providing precise information between the variables/wavelengths and observations/concentration, which is otherwise not possible, as spectral data consists of a large volume of variables [23]. Partial Least Square Analysis followed by leave-one-out cross validation has been used to investigate the signature wavelengths of lactate and the entire ranges of the obtained NIR spectra. This study will demonstrate the capabilities and differences of NIR dual beam and FTNIR spectroscopy coupled with multivariate analysis for lactate concentration determination.

## 2. Reagents and Materials

Na-L-Lactate powder and isotonic buffer (PBS) (Analytical Grade) were obtained from Thermo Fisher Scientific (Waltham, MA, USA) and a 600 mM stock solution of lactate was prepared. Forty one sample solutions of varying lactate concentration, at intervals of 0.5 mM), of 30 mL each were prepared by diluting the stock solution. All the solutions were maintained at a room temperature of $24{\;}^{\circ }$C at a pH 7.4.

Sheep whole blood in Alsever’s solution was acquired from TCS Biosciences Ltd. (Buckingham, UK). Sheep blood was the preferred choice for this study because its lactate concentrations provide a close match to normal and temporal changes in lactate levels in human blood. The study was approved by the Senate Research Ethics Committee (SREC), City, University of London (SREC 17-18 05 6ii 27.06.2018).

In total, 1 mL of lactate in PBS solutions was then mixed with 19 mL of sheep blood to obtain 41 samples (for each set) of varying lactate concentrations of 20 mL each at 0.5 mM concentration intervals. The initial lactate concentrations of the commercially acquired sheep blood were 3.8–4.0 mM. This method of sample preparation was similar to a previous study [24]. The concentrations of lactate and the pH of the solution samples were measured using an ABL 825 blood-gas analyser from Radiometer UK Limited (Crawley, UK). Lactate measurement, here, is based on an electrochemical lactate biosensor that uses the enzyme lactate oxidase to produce detectable $H_{2}O_{2}$; values were found to be 4.5 to 13.8 mM with intervals of 0.5 mM and pH 7.1–7.4, respectively. These samples were then used directly in the spectrometers without further treatment.

## 3. NIR Spectrometry

### 3.1. Dual Beam Dispersive NIR

The spectra of the prepared samples, were collected using a Lambda 1050 dual beam spectrophotometer from Perkin Elmer Corp. (Waltham, MA, USA). A 100 mm InGaAs integrating sphere detector set at 0 deg transmission mode was used as detector and spectra were acquired in the wavelength range 870–2600 nm at 2 nm intervals. The integrating sphere detector helped to achieve a homogenised/reduced scatter transmitted light through the whole blood samples. The Gain and Response Times for the detector was maintained at 0 and 0.2 s, while slit size was kept at 2 nm. These settings prevented the detectors from oversaturation. Reference beam attenuation was set at 1%, and quartz cuvettes of 1 mm path length from Hellma Materials GmbH (Jena, Germany) were used. A Spectralon Diffuse Reflectance Standard from Lapsphere (North Sutton, NH, USA) was also placed at the aperture of the sphere detector for both baseline corrections and sample spectra collections in order to close the sphere. Three spectra were collected from each sample, which were then averaged to obtain a 41 spectra data-set. In this technique, one spectrum is produced per scan and the time of acquisition was on average 5 min per spectrum.

### 3.2. FTNIR

Spectra from 900–3000 nm, were obtained using a Frontier${}^{\mathrm{TM}}$ FTIR/NIR spectrometer, again from Perkin Elmer, (Waltham, MA, USA) for the same whole blood samples, simultaneously to avoid the effects of any temporal sample change. Spectra were collected at 0.8 nm data intervals, spectral resolution maintained at 2 cm${}^{-1}$ and the variable J-stop (Jacquinot-stop) was kept at 3.16 mM at 3000 cm${}^{-1}$, for a better throughput.

Multiplexing of 20 scans per sample, with a scan speed of 0.2 cm s${}^{-1}$ was adopted, in order to improve SNR. 70 μL of each sample was then pipetted directly onto the ZnSe crystal surface of the Horizontal Attenuated Total Reflectance (HATR) accessory, from Pike Technologies Inc., (Fitchburg, WI, USA). The specifications of the HATR accessory were: a trough Zinc Selenide (ZnSe) crystal of 4 mM thickness, the length and refractive index of which were 80 mM and 2.4, respectively. The crystal allowed 10 internal reflections per measurement. The time of acquisition in this instrument was on average 5 min for all 20 scans.

Background scans with the empty crystal for ambient condition deductions were taken after either 2 samples or every 20 min during spectra collection. Thus, a total of 41 spectra were obtained, which were then pre-processed and analysed.

### 3.3. Spectral Analysis

Spectra collection and visualisation was carried out using the software packages:UVWin Lab for LAMBDA 1050, from Perkin Elmer (Waltham, MA, USA) andSpectrum 10 for Frontier${}^{\mathrm{TM}}$ FTIR/NIR, from Perkin Elmer (Waltham, MA, USA).

Further pre-processing of the spectra was in MATLAB R2020b, MathWorks${}^{\mathrm{TM}}$ (Natick, MA, USA). The following steps were performed on the spectra for both the data-sets separately, in succession [25]:Spectral Difference: The spectrum of the first sample (4.2 mM concentration of lactate) was subtracted from the rest of the spectraLinear Robust Multiplicative Scatter Correction (MSC) [26] was used for both the data-setsSavitzky-Golay (SG) [27] filter parameters (Polynomial Order, Derivative and Window Length) for both the data-sets are mentioned in Table 1. These parameters (smoothing, derivative, polynomial degree) were selected based on expert knowledge/rule of thumb/visual investigation.

The spectra were then analysed using the principles of Aquaphotomics [28], Linear Regression on observed peaks relevant to lactate [24] and Partial Least Square Regression (PLSR) [29] for predictive modelling. All these models and analysis was investigated with the help of algorithms developed in house in MATLAB.

These models were built up using Latent Variables (LVs), the optimum number of which were calculated from Predicted Residual Sum of Squares (PRESS) results for both the instrumental data-sets and considered as the saddle point [30]. The models were then validated using the leave-one-out cross validation technique. In this process each spectrum is left out and predicted using the N-1 (N = 41, in this case) spectra. This process is repeated N times such that each spectrum is left out and predicted using the others to obtain the calibration model. Using this process, predictions were compared with the actual concentrations of lactate obtained using the blood-gas analyser, the gold standard for lactate measurement. The Coefficient of Determination ($R^{2}$), and the Root Mean Squared Error of Cross Validation (RMSECV) the were finally used to determine accuracy of the models and compared against each other.

## 4. Results and Discussion

In order to absorb infrared radiation, a molecule must have a non-zero dynamic dipole moment. This favours vibrational transitions to the discrete energy states n = 0 to n = ±1,±2,±3, …, where n is the principal quantum number. A transition in the rotational quantum number of a molecule, J can also be induced simultaneously in the near infrared region of the electromagnetic spectrum. Thus, the NIR absorption spectrum comprises of various vibrational-rotational transition nodes. Besides these fundamental modes of vibration, peaks also arise from transition combinations (two or more fundamental vibrations becoming excited simultaneously) and overtones (first overtone, n = 0 to n = 2 or second overtone, n = 0 to n = 3, and so on) bands. These special vibration bands are only seen in the NIR spectral region. However, in the spectra collected during this study, the peaks arising from transitions of the lactate molecule were overshadowed by the overtones for water -OH bonds, and therefore, spectra obtained using both types of instruments strongly resembled typical NIR spectra of any aqueous solution [31]. The figures for both data-sets are shown in Figure 1. Hence, in order to extract the peaks relevant to lactate, both spectral data sets were divided into three regions, and are discussed below.

### 4.1. Spectral Regions with O-H Absorption Interference

The wavelengths of interest which depict all the -OH functional groups are: 970 nm and 1820 nm (O-H stretch), 1450 nm, 1490 nm and 1540 nm (O-H stretch first overtone), 1960 nm (O-H stretch/O-H bend combination), 2070 nm (O-H combination) and 2100 nm (O-H bend) [32]. Table 2 shows results of the linear regression, which was performed in order to understand the effects of lactate concentration changes in absorbance at these wavelengths of interest. The coefficients and standard errors (in parenthesis) are presented, along with statistical significance marker *p*-values ≤ 0.05.

From Table 2, it can be seen that at these wavelengths, the absorption values are not linearly correlated at a high level of statistical significance. For further understanding of the effects of O-H in the spectra, this region was analysed using the principles of aquaphotomics [33]. This principle states that in an aqueous medium, water-light interaction, at the NIR overtone regions of water could reveal indirect information that relate and are pertinent to the overall system. This is called the “water mirror approach” and utilizes the H bonds of water that form covalent bonds with the analytes present in solution and which are usually disturbed by minute perturbations in the system.

Aquagrams depicting activated Water Absorbance Spectral Patterns (WASPs) were constructed using 12 Water Matrix Absorbance Coordinates (WAMACs) (1342 nm (1344 nm for FTNIR), 1364 nm, 1374 nm (1372 nm for FTNIR), 1384 nm, 1412 nm, 1426 nm (1424 nm, for FTNIR), 1440 nm, 1452 nm, 1460 nm, 1476 nm, 1488 nm and 1512 nm) in the first Water Absorbance Bands (WABS) (1300–1600 nm). Figure 2, depicts the aquagrams for both the NIR data sets.

Visual inspection reveals that the aquagram obtained from the dual beam dispersive NIR instrument was clearer with sharper lines and followed patterns towards particular wavelengths, in comparison to FTNIR collected spectra. This may be due to the better SNR instrument capabilities of the former type. It could also be seen that the WAMACs that were activated for both the aquagrams were: 1412 nm ($C_{5}$), 1426 nm ($C_{6}$) and 1440 nm ($C_{7}$), which are smaller in wavelength. Previous studies have shown that the chemical/molecular conformations that could be assigned to these WAMACs are the following: $C_{5}$: water with free hydroxyl OH${}^{-}$ side groups, $C_{6}$: H-OH bend (δ), $C_{7}$: Water molecules with 1 H bond [34,35]. This was apparent as the solutions had a substantial amount of water and these free water molecules were highlighted by the shorter wavelength WAMACs [33].

Apart from these, a few longer wavelengths, 1476 nm, 1488 nm and 1512 nm, are seen to be activated as well. These wavelengths could be assigned the following WAMACS and chemical/molecular conformations: $C_{10}$: Water molecules with 2 H bonds, $C_{11}$: Water molecules with 3 H bonds, $C_{12}$: Strongly bound water. These water conformations in the whole blood aqueous system may be the result of water-lactate molecular interactions and the formation of ionic lactate complex clusters, as whole blood is a polar solvent that can sustain ionic complexes [33]. It is also possible that bonded lactate structures in aqueous solution were the result of lactate ions bound to plasma proteins [36] or to deoxy- and oxyhaemoglobin [37]. Because of these bound complexes, the effects of concentration change of different free analytes would not be seen in such an aqueous system [38]. Hence, when PLS predictive models were built using this spectral region (1300–1600 nm), using 4 and 2 LVs for the dual beam dispersive NIR and FTNIR data-sets respectively; the $R^{2}$ and RMSECV values were 0.51, 0.54 and 1.28 mM and 1.83 mM respectively. Thus, confirming that neither of these data-sets could be used for lactate concentration prediction in this spectral region of the NIR.

### 4.2. Signature Wavelengths of Lactate

Previous studies have shown that the NIR spectral region is pH dependent and the wavelengths that are pertinent to lactate concentration changes depend on the pH of the solution in which they are measured [32,39]. Since, the pH calculated for these sample solutions were between 7.1 and 7.4, the wavelengths of interest were: 1142 nm, 1232 nm, 1280 nm, 1330 nm, 1710 nm, 1750 nm, 1882 nm, 2204 nm, 2320 nm and 2340 nm [32]. Linear Regression on these wavelengths are shown in Table 3. Moreover, a few more wavelengths can be found in the literature where studies similar to this were performed for lactate concentration measurements at similar pH: 923 nm and 1047 nm [40], 1675 nm, 1690 nm and 1730 nm [41], 2166 nm, 2254 nm and 2292 nm [42]. Linear regression on these wavelengths was performed separately, shown in Table 4. (Since the spectra for the Lambda 1050 and the Frontier FTIR/NIR was collected at 2 nm and 0.8 nm data intervals, hence the wavelengths available closest were considered for analysis.)

Of all the wavelengths stated above, the ones which were statistically significant are: 1046 nm (*p*-values ≤ 0.00005), 1142 nm (*p*-values ≤ 0.00005), 1232 nm (*p*-values ≤ 0.00005), 1280 nm (*p*-values ≤ 0.05), 1330 nm (*p*-values ≤ 0.05), 1690 nm (*p*-values ≤ 0.00005), 1710 nm (*p*-values ≤ 0.005), 1750 nm (*p*-values ≤ 0.005), 2166 nm (*p*-values ≤ 0.05), 2320 nm (*p*-values ≤ 0.00005) and 2340 nm (*p*-values ≤ 0.05). While for the FTNIR data set, none of the wavelengths showed any linear statistical significance correlation. This might again be due to better SNR of the dual beam dispersive compared to the FTNIR instrument for each wavelength.

However, when these particular wavelengths were used for lactate concentration prediction using partial least square algorithm, the $R^{2}$ and RMSECV values were 0.68 and 1.45 mM respectively, for the dual beam dispersive data set (using 9 LVs). Similarly, for the FTNIR data set(using 8 LVs), the $R^{2}$ and the RMSECV values were 0.78 and 2.61 mM respectively. These results show that using the wavelengths pertinent to lactate in the NIR spectral region, the concentration of lactate can be predicted. The FTNIR instrument gives better prediction results compared to the dispersive type. The reason behind the improvement of results may be due to the use of multivariate chemometric techniques, which is a better analysis tool for NIR spectra since band identification is a demanding process in NIR spectroscopy due to the complexity of peaks that are often the result of overtones and combination of -CH and -OH bands.

### 4.3. Full Spectral Regions for Predictive Modelling

Lastly, PLS predictive models were constructed for both the data sets using the full NIR spectral region (800–2300 nm). The LVs used to create the simplest models for each data set were 9 and 6 for the dual beam dispersive NIR data-set and the FTNIR data set, respectively.

Cross-validation of these models was performed by leave-one-out on individual data-sets, as shown in Figure 3. The coefficients of determination ($R^{2}$) of each set, were 0.75 for dual beam dispersive NIR and 0.85 for the FTNIR set. The RMSECV for the dual beam dispersive NIR set was 1.23 mM and for the FTNIR set was 1.02 mM. Hence, it can be seen that the FTNIR instrument is better than its counterpart in predicting lactate concentrations when the full NIR spectral range is considered for prediction.

From the two sets of results presented here, it is evident that FTNIR is a better instrument for the accurate determination/prediction of lactate concentrations. A few possible advantages of the FTNIR over the dual beam dispersive NIR instrument are: speed (spectra in an FTNIR can be obtained faster because the full range of wavelengths are scanned at one step), better electronics in the instrument with fewer moving parts and higher spectral resolution [13,14]. For the same reasons, the FTNIR instrument is believed to have provided better results, as compared to its counterpart.

## 5. Conclusions

It is important to understand the instrument/technological capabilities of a dual beam dispersive spectrophotometer, compared to an FTNIR spectrometer for biomedical applications, as both of them have definite advantages. Hence, in this study, lactate concentration changes into the different spectral regions within the NIR were studied. It could be seen that for the regions of the NIR which are influenced by -OH absorption, lactate could not be predicted with a high accuracy using either instrument. However, for lactate predictive analysis using only the particular wavelengths which serve as ’signatures’ for lactate in the NIR region at pH 7 and 7.4, the accuracy of prediction was increased by 17% for the dual beam dispersive spectrophotometer and by 24% for the FTNIR spectrometer. Finally, when the full range of NIR was taken into consideration, prediction capability further increased by 7% for both the spectrometers. This might be because the NIR spectrum as a whole provides a fingerprint of the molecular environment. Overall, lactate concentration can be predicted with better accuracy using FTNIR compared with dual beam dispersive spectrophotometer for all spectral regions.

Technological advances for portable NIR instrumentation has gained focus recently, for example, portable NIR spectrometers are now equipped with solid state imaging detectors in compact integrating circuits (ICs), which can, for example, collect as many as 30,000 spectra per second, thus massively improving speed of spectra acquisition [43,44]. For a portable FTNIR, like the Neo-Spectra by Si-Ware Systems (Cairo, Egypt) and NANOQuest-2.5 by OceanOptics (Largo, FL, USA) Micro Electromechanical Systems (MEMS)-based Michelson interferometers are now made in a single chip, which has all the advantages of the FTNIR as compared to its counterpart. Similar studies performed using such novel portable NIR spectrometers for various biochemical applications will further help advance non-invasive instrumentation of the clinical environment.

## Figures and Tables

**Figure 1 sensors-21-01891-f001:**
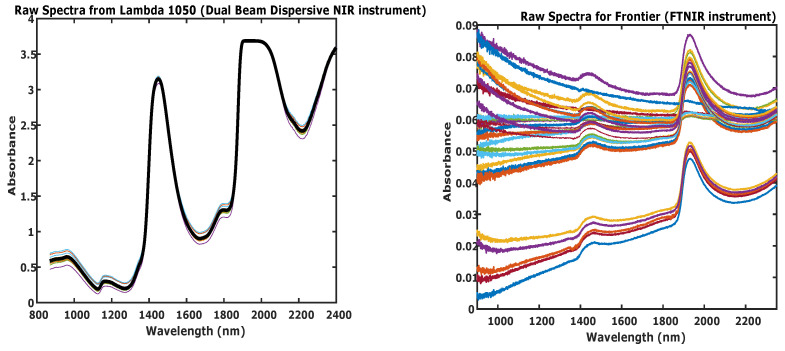
Raw Spectra of the 41 samples (different concentrations) of lactate for (**left**) Lambda 1050 (dual beam dispersive NIR instrument) and (**right**) Frontier (FTNIR instrument). The different coloured spectra depict the concentrations of lactate from 4.5 to 13.8 mM at 0.5 mM intervals.

**Figure 2 sensors-21-01891-f002:**
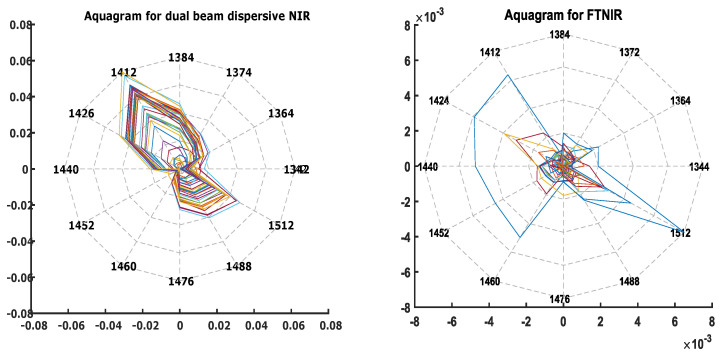
Aquagrams for lactate in whole blood samples in Water Absorbance Bands (WAB) 1300–1600 nm, showing Water Absorbance Spectral Pattern (WAPS) in the Water matrix co-ordinates (WAMACS) of varying concentrations of lactate in PBS samples. The WAMACS: $C_{1}$: 1342 (1344), $C_{2}$: 1364, $C_{3}$: 1374 (1372), $C_{4}$: 1384, $C_{5}$: 1412, $C_{6}$: 1426 (1424), $C_{7}$: 1440, $C_{8}$: 1452, $C_{9}$: 1460, $C_{1}0$: 1476, $C_{11}$: 1488 and $C_{12}$: 1512 nm depicts molecular conformations which arises due to water-NIR light-lactate molecule interactions. The Aquagram on the left is constructed from the dual beam dispersive NIR spectra data-set and the one on the right is constructed from FTNIR spectra data-set.

**Figure 3 sensors-21-01891-f003:**
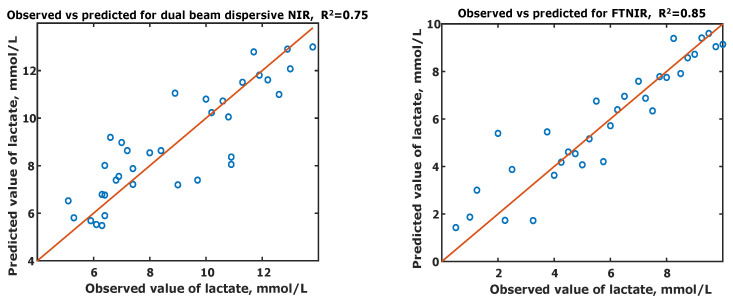
Observed (known) lactate concentration versus the Predicted concentrations of forty-one samples of lactate in whole blood. The correlation coefficient ($R^{2}$) and the Root Mean Squared Error of Cross Validation (RMSECV) for each set are also shown. (**left**) The Coefficient of Determination ($R^{2}$) and the Root Mean Squared Error of Cross Validation (RMSECV) for dual beam dispersive NIR data-set are 0.75 and 1.23 mmols/L respectively; (**right**) The Coefficient of Determination ($R^{2}$) and the Root Mean Squared Error of Cross Validation (RMSECV) for FTNIR set are 0.85 and 1.02 mmols/L respectively.

**Table 1 sensors-21-01891-t001:** Parameters for Savitzky-Golay (SG) Filter.

Data-Set	Polynomial Order (PO)	Derivative (D)	Window Length (WL)
Dual Beam
Dispersive NIR	2	1	7
FTNIR	2	1	31

**Table 2 sensors-21-01891-t002:** Linear Regression on observed peaks (nm) arising in the spectra due to O-H bonds. (coefficient and Standard errors in parenthesis. *p*-values depict statistical significance markers ≤ 0.05).

Wavelength (nm)/Data-Set	Dual Beam Dispersive NIR	FTNIR
970	1.51 × $10^{-5}$ (9.15 × $10^{-6}$)	−2.00 × $10^{-6}$ (1.43 × $10^{-6}$)
1450	−1.11 × $10^{-4}$ (4.17 × $10^{-5}$) ^1^	−1.72 × $10^{-7}$ (6.8 × $10^{-7}$)
1490	−1.16 × $10^{-5}$ (3.56 × $10^{-5}$)	−1.28 × $10^{-6}$ (5.42 × $10^{-7}$) ^1^
1540	2.28 × $10^{-5}$ (1.17 × $10^{-5}$)	−4.94 × $10^{-7}$ (5.26 × $10^{-7}$)
1820	3.73 × $10^{-5}$ (1.46 × $10^{-5}$) ^1^	1.33 × $10^{-7}$ (3.57 × $10^{-7}$)
1960	4.25 × $10^{-6}$ (4.42 × $10^{-6}$)	−5.97 × $10^{-6}$ (3.03 × $10^{-6}$)
2070	2.77 × $10^{-5}$ (3.79 × $10^{-5}$)	−1.72 × $10^{-6}$ (9.26 × $10^{-7}$)
2100	7.51 × $10^{-5}$ (3.41 × $10^{-5}$) ^1^	−1.62 × $10^{-6}$ (6.59 × $10^{-7}$) ^1^

^1^*p*-values ≤ 0.05.

**Table 3 sensors-21-01891-t003:** Linear Regression on wavelengths pertinent to lactate in both the data-sets.

Wavelength (nm)/Data-Set	Dual Beam Dispersive NIR	FTNIR
1142	1.08 × $10^{-4}$ (1.27 × $10^{-5}$) ^3^	−6.71 × $10^{-7}$ (1.01 × $10^{-6}$)
1232	−5.26 × $10^{-5}$ (9.28 × $10^{-6}$) ^3^	−7.30 × $10^{-7}$ (6.72 × $10^{-7}$)
1280	1.08 × $10^{-5}$ (4.85 × $10^{-6}$) ^1^	−5.72 × $10^{-7}$ (6.86 × $10^{-7}$)
1330	4.29 × $10^{-5}$ (1.69 × $10^{-5}$) ^1^	−5.84 × $10^{-7}$ (7.01 × $10^{-7}$)
1710	−5.34 × $10^{-5}$ (1.51 × $10^{-5}$) ^2^	3.64 × $10^{-7}$ (4.88 × $10^{-7}$)
1750	3.03 × $10^{-5}$ (1.09 × $10^{-5}$) ^2^	7.04 × $10^{-7}$ (4.62 × $10^{-7}$)
1882	−2.18 × $10^{-4}$ (0.09 × $10^{-4}$)	1.46 × $10^{-5}$ (7.50 × $10^{-6}$)
2204	2.19 × $10^{-5}$ (2.38 × $10^{-5}$)	3.12 × $10^{-7}$ (3.15 × $10^{-7}$)
2320	−1.26 × $10^{-4}$ (3.26 × $10^{-5}$) ^3^	8.18 × $10^{-7}$ (8.18 × $10^{-7}$)
2340	1.3833 × $10^{-4}$ (6.07 × $10^{-5}$) ^1^	2.41 × $10^{-6}$ (9.93 × $10^{-7}$)

^1^*p*-values ≤ 0.05. ^2^
*p*-values ≤ 0.005. ^3^
*p*-values ≤ 0.00005.

**Table 4 sensors-21-01891-t004:** Linear Regression on wavelengths pertinent to lactate in both the data-sets.

Wavelength (nm)/Data-Set	Dual Beam Dispersive NIR	FTNIR
922	1.354 × $10^{-5}$ (9.59 × $10^{-6}$)	−3.42 × $10^{-6}$ (1.73 × $10^{-6}$)
1046	5.57 × $10^{-5}$ (1.52 × $10^{-5}$)	−1.67 × $10^{-6}$ (1.30 × $10^{-6}$)
1674	−1.63 × $10^{-6}$ (2.84 × $10^{-5}$)	1.47 × $10^{-9}$ (5.47 × $10^{-7}$)
1690	3.72 × $10^{-5}$ (9.10 × $10^{-6}$)	−1.05 × $10^{-8}$ (3.65 × $10^{-7}$)
1730	4.29 × $10^{-6}$ (1.10 × $10^{-5}$)	−1.59 × $10^{-7}$ (5.44 × $10^{-7}$)
2166	−6.09 × $10^{-5}$ (2.66 × $10^{-5}$)	−2.52 × $10^{-7}$ (2.96 × $10^{-7}$)
2254	3.72 × $10^{-5}$ (3.04 × $10^{-5}$)	5.85 × $10^{-7}$ (7.17 × $10^{-7}$)
2292	−3.36 × $10^{-5}$ (4.69 × $10^{-5}$)	1.77 × $10^{-6}$ (5.60 × $10^{-7}$)

## Data Availability

The data presented in this study are available on request from the corresponding author.

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
