# Peer review of "Comparison of Dual Beam Dispersive and FTNIR Spectroscopy for Lactate Detection"

_sensors, 2021, doi:10.3390/s21051891_

Round 1
Reviewer 1 Report
This study describes the comparison of two different IR systems (dual beam dispersive and FTNIR) for the prediction of lactate concentration in blood, in a clinically- relevant range.
This is not a ground-breaking study and can be barely considered novel. The authors conclude that FTNIR showed 10 % better predictive capability compared to the dual beam dispersive NIR spectrometer when taking into account the whole spectrum. However they also state that for FTNIR none of the wavelengths showed any linear statistical significance correlation and that this ‘’might be due to better SNR of the dual beam dispersive compared to the FTNIR instrument.’’
As a result no physically meaningful spectral bands which are responsible for the lactate concentration prediction are determined. The authors should provide a better explanation on where they think the (full-spectrum) prediction is based.
Additionally, P-values in Table 1 and 2 are not obvious. A more descriptive legend should be added (similarly to the one in the main body ‘’.. the coefficients and standard errors (in parenthesis) are presented, along with statistical significance marker p-values ≤ 0.05..’’
Author Response
Attached as a word file

Reviewer 2 Report
Dear authors. The main suggestions can be seen in the attached "letter to authors".

Author Response
Attached as a word file

Round 2
Author Response
We would like to thank the reviewer again for their time.
Reviewer 2 Report
Just a comment.
I want the authors to insert the percentages of samples chosen for calibration and prediction of PLS models in the Material and Methods. Whether the samples chosen for lactate prediction were internal or external? (if it was done). Otherwise, the authors should mention how the PLS calibration models were built, whether they were only calibrated or what approach was used by the authors.
General comment
In the future, I advise authors to evaluate the use of sample selection algorithms, such as Kennard-Stone, SPXY or others. (Again, if this approach has not been used in the present work). Commonly in my chemometrics works, colleagues and I use the Kennard-Stone or SPXY algorithms to choose the calibration and prediction samples, in a percentage of 67% for calibration and 33% for prediction, commonly used in PLS models for calibrated and validation.
Author Response
We would like to thank the reviewer again for his comment and suggestion. We would consider using sample selection algorithms for future work. However, for now the following line has been added in the manuscript: "In this process each spectrum is left out and predicted using the N-1 (N=41, in this case) spectra. This process is repeated N times such that each spectrum is left out and predicted using the others to obtain the calibration model."